# Usefulness of Imaging and Biological Tools for the Characterization of Portal Vein Thrombosis in Hepatocellular Carcinoma

**DOI:** 10.3390/diagnostics12051145

**Published:** 2022-05-05

**Authors:** Călin Burciu, Roxana Șirli, Felix Bende, Renata Fofiu, Alina Popescu, Ioan Sporea, Ana-Maria Ghiuchici, Bogdan Miuțescu, Mirela Dănilă

**Affiliations:** 1Department of Gastroenterology and Hepatology, “Victor Babeș” University of Medicine and Pharmacy, 300041 Timișoara, Romania; burciu.calin@gmail.com (C.B.); bendefelix@gmail.com (F.B.); alinamircea.popescu@gmail.com (A.P.); isporea@umft.ro (I.S.); anamaria.ghiuchici@gmail.com (A.-M.G.); bmiutescu@yahoo.com (B.M.); mireladanila@gmail.com (M.D.); 2Advanced Regional Research Center in Gastroenterology and Hepatology, “Victor Babeș” University of Medicine and Pharmacy, 30041 Timișoara, Romania; renata.fofiu@yahoo.com

**Keywords:** CEUS, portal vein thrombosis, alpha-fetoprotein, hepatocellular carcinoma, tumor in vein

## Abstract

This study aims to evaluate the performance of contrast-enhanced ultrasound (CEUS) and biological tests to characterize portal vein thrombosis (PVT). We retrospectively analyzed 101 patients with PVT, liver cirrhosis, and hepatocellular carcinoma (HCC). In all patients, demographic, biologic, imaging, and endoscopic data were collected. All patients underwent CEUS and a second line imaging technique (CE-CT/MRI) to characterize PVT. Of the 101 cirrhotic subjects, 77 (76.2%) had HCC. CEUS had 98.6% sensitivity (Se) and 89.3% specificity (Sp) for the characterization of PVT type. A significant correlation was found between alpha-fetoprotein (AFP) levels and the PVT characterization at CEUS (r = 0.28, *p* = 0.0098) and CT/MRI (r = 0.3, *p* = 0.0057). Using the AFP rule-out cutoff values for HCC (AFP < 20 ng/dL), 78% of the subjects were correctly classified as having benign PVT, while 100% of the subjects were correctly classified as tumor-in-vein (TIV) when the rule-in cutoff value was used (AFP ≥ 200 ng/dL). Using multiple regression analysis, we obtained a score for classifying PVT. The PVT score performed better than CEUS (AUC—0.99 vs. AUC—0.93, *p* = 0.025) or AFP serum levels (AUC—0.99 vs. AUC—0.96, *p* = 0.047) for characterizing PVT. In conclusion, CEUS is a sensitive method for the characterization of PVT. The PVT score had the highest performance for PVT characterization.

## 1. Introduction

The most significant liver vascular disorder is portal vein thrombosis (PVT) [1]. PVT occurs due to the formation of a thrombus within the portal vein (PV) and its emerging branches and is a redoubtable complication of liver cirrhosis [2].

According to Ogren’s study that included 23,796 consecutive autopsies, 23% of PVT were secondary to a primary hepatobiliary cancer, 44% to a malignancy of the hepatobiliary region, and 28% of cases were secondary to liver cirrhosis [3]. 

In patients with compensated cirrhosis, PVT occurs in less than 1% of cases, compared with 8–25% in patients with indications for liver transplantation [4,5]. Approximately 40% of patients with hepatocellular carcinoma (HCC) can develop PVT, a severe complication that influences prognosis and significantly reduces the therapeutic options [6], being a contraindication for liver transplantation [7].

In patients with HCC, PVT is caused by the invasion of the portal system by the malignant tumor, also known as tumor-in-vein (TIV). In cirrhotic patients without HCC, a mix of local and systemic factors are involved in PVT development: hyper-coagulopathy, venous stasis, and endothelial injury [8]. 

The diagnosis of PVT begins with B-mode ultrasound (US) examination, where the PV can appear normal (anechoic) or with a hypoechoic mass inside, in which case a PVT is suspected. In this case, color and spectral Doppler should be performed. Complete blunt PVT is associated with the absence of a Doppler signal, but the presence of an intrathrombus signal is highly specific for malignancy, even if with low sensitivity [9]. 

Contrast-enhanced ultrasound (CEUS) is a helpful tool for PVT characterization. Benign PVT is avascular in all phases of CEUS, while TIV usually has the same enhancement pattern as the primary tumor, with arterial hyperenhancement followed by washout in the portal and late phases. CEUS is a valid method for evaluating PV permeability, for characterization of thrombus extension, and for differentiation of the type of thrombosis, benign vs. malignant [10]. Furthermore, CEUS has been proven to be a helpful tool for diagnosing and characterizing HCC and other focal liver lesions (FLL) [11,12] and also for nonhepatic applications [13], such as the identification and characterization of pancreatic cystic neoplasms [14].

Numerous guidelines combine biomarkers with ultrasound examination to diagnose HCC and its complications [15,16,17]. An intensively tested serological marker is alpha-fetoprotein (AFP), used in surveillance, diagnosis, and treatment response of HCC. Increased levels of AFP are a risk factor for HCC development. The HCC size and volume at the time of diagnosis are correlated with AFP levels [15]. However, a large study in more than 1700 patients with HCC [18] had shown that a significant part of the study cohort had AFP levels below the recommended cutoff for surveillance programs (<20 ng/mL) [17]. Furthermore, if the HCCs were smaller than 5 cm, in approximately half of the cases, the AFP was lower than the recommended cutoff for surveillance [18]. The AFP levels increase in patients with active hepatitis or cirrhosis, while low levels of AFP are correlated with a good response to nucleos(t)ide analogs in chronic hepatitis B [17] or to direct-acting antivirals (DAA) treatment for chronic hepatitis C [19]. It was also noticed that AFP levels decrease in patients with HCC after treatment [20]. AFP has been primarily tested in diagnosing HCC rather than for surveillance, with studies showing only an additional 6–8% detection rate when combined with US examination [21]. AFP levels are typically elevated in HCC patients with PVT, but whether they are cause or consequence is unclear [22].

This study aims to evaluate the sensitivity (Se) and specificity (Sp) of CEUS for the characterization of the PVT as benign vs. malignant using contrast-enhanced (CE) computer-tomography (CT) or CE magnetic resonance imaging (MRI) as reference diagnostic methods. The association between AFP levels and PVT type in cirrhotic patients with and without HCC was also evaluated.

## 2. Materials and Methods

### 2.1. Patients

We retrospectively analyzed 101 consecutive patients with macroscopic PVT secondary to liver cirrhosis with or without HCC that were admitted between January 2016 and December 2020 to a tertiary gastroenterology and hepatology center. 

The inclusion criteria were: patients with cirrhosis with or without HCC, with PVT, recently diagnosed using B-mode ultrasound, and subsequently characterized by CEUS. Another necessary criterion was to perform a CE-CT or CE-MRI to characterize PVT. The study did not include patients with known PVT and those unable to undergo the reference diagnostic method to characterize the type of thrombosis. 

The diagnosis of liver cirrhosis and HCC was established using the European Association for the Study of the Liver (EASL) guidelines’ criteria [15]. In all patients, data were collected in a database, including age, gender, etiology of cirrhosis, Child–Pugh Score, Model for End-Stage Liver Disease (Meld) Score, the extension of PVT, serum AFP, the presence of esophageal and gastric varices, and HCC maximum tumor diameter. 

All patients underwent CEUS and a second line imaging technique (CE-CT/MRI), where PVT has been characterized according to the enhancement pattern following contrast. 

Upper digestive endoscopy was performed on 84/101 subjects, and we characterized the grade of varices according to the Baveno classification for esophageal varices [23] and Sarin classification for gastric varices [24].

### 2.2. Determination of AFP Values

The AFP values were determined from venous blood; the samples (5 mL of venous blood) were collected from fasting patients in a vacutainer without anticoagulants. After the blood clot formed at room temperature, the serum was separated by centrifugation in the first 4 h after collection. The minimum amount of serum required was 0.5 mL. Subsequently, AFP was quantified in the resulting serum using a VITROS^®^ XT 7600 or a VITROS^®^ XT 3600 device, with a detection limit of 0.24 ng/mL.

Of the 101 patients included in the study, 81 had a recorded serum AFP and were analyzed separately regarding the correlation between AFP and PVT type. We used two AFP level cutoffs for the diagnosis of TIV on the basis of EASL HCC management guidelines [15]: 20 ng/mL and 200 ng/mL.

### 2.3. Diagnostic of PVT

We first suspected the diagnosis of PVT by the B-mode US performed using high-resolution ultrasound machines: GE Healthcare Logic E9 and Philips Epiq 7, based on echogenicity in the lumen of the PV. Experts in ultrasound and CEUS performed all examinations.

Doppler examination has been performed to identify a detectable signal inside the PV. CEUS was the first contrast imaging method used to investigate the nature of PVT (benign vs. malignant). According to the European Federation of Societies for Ultrasound in Medicine and Biology (EFSUMB) guidelines [11], the thrombus was considered benign if it was avascular (no enhancement in any vascular phase of CEUS) (Figure 1), as opposed to TIV (hyperenhancement in the arterial phase with “washout” in the portal or late phases) (Figure 2). 

In our study, CEUS was performed using SonoVue^®^ (Bracco, Italy) as a contrast agent; 2.4 mL of SonoVue^®^ were injected intravenously into the left forearm, for each examination. All three US examinations, B-mode US, Color Doppler, and CEUS, were performed by the same operator on the same machine in the same session. CEUS images were interpreted in real time by an expert in the field. Furthermore, all CEUS images have been recorded and stored to be available for reassessment.

### 2.4. Diagnostic Reference Method

All cases underwent CE-CT (*n*= 82) or CT-MRI (*n* = 19) in our study. During CT assessment, the evaluation began with a native phase, followed by mandatory dynamic evaluation with contrast, which included three phases. The first phase—Arterial (late arterial: 30–35 s. from the beginning of contrast injection); the second phase—Portal (60–70 s); and the third phase—Parenchymal (3–5 min). In each patient, 1–1.5 mL/kg iodinated contrast was injected by an automated syringe with a 3 mL/s flow rate.

Similar to the CT evaluation, MRI began with a native sequence followed by a contrast in dynamics. Because of the nature of our cohort, which included patients with HCC, we used two types of contrast: extracellular contrast (Gadovist) or hepatobiliary contrast with dual extracellular and intracellular behavior (Gd-EOB-DTPA-Primovist). The contrast agent injection was performed with an automatic syringe with a flow rate of 1–2 mL/s. The doses were correlated with the type of contrast agent used. The contrast sequence included three phases: arterial phase (late arterial < 30 s from the injection), porto-venous phase (60–70 s), and extracellular phase (equilibrium) > 120 s. When a hepatobiliary contrast agent was used, another phase was added—the hepatospecific phase (2 h after the contrast bolus for Gadovist and 20 min for Primovist). All images were recorded and stored. Experienced radiologists characterized the type and extension of PVT based on CE-CT or CE-MRI imaging.

### 2.5. Extension of PVT

We classified the thrombus extension into four grades: Grade 1: Partial PVT–obstruction of less than 50% of the PV lumen; Grade 2: more than 50% or complete occlusion of the PV, with or without minimal extension into the superior mesenteric vein (SMV); Grade 3: Complete thrombosis of the PV with thrombus extension to the proximal part of the SMV; Grade 4: complete thrombosis of the PV with thrombus extension to the proximal and distal part of the SMV [25].

### 2.6. Statistical Analysis

The statistical analysis was performed using MedCalc Version 19.4 (MedCalc program, Belgium) and Microsoft Office Excel 2019 (Microsoft for Windows). Descriptive statistics were used for demographic and laboratory findings. The Kolmogorov–Smirnov test was used to establish the distribution of numerical variables. Numerical variables with normal distribution were presented as means ± standard deviation, while variables with non-normal distribution were presented as median values and range. 

The differences between groups were assessed using the Student’s *t*-test for continuous variables with normal distribution, while the Mann–Whitney *U* test was used for continuous variables without normal distribution. 

The Fisher’s and Pearson’s chi-squared tests were used to compare proportions. Se, Sp, positive predictive value (PPV), and negative predictive value (NPV) of CEUS for characterization of PVT were calculated using CT/MRI as the reference method, and 95% confidence intervals (CI) were calculated for each predictive test; a *p*-value below 0.05 was considered to concede statistical significance.

Univariate regression analysis and multivariate regression analysis were used to identify the factors associated with the presence of malignant PVT and to formulate the score for malignant PVT prediction. Areas under receiver operating characteristic (AUROC) curves were calculated for the PVT score values to identify the discriminating cutoff value for the score. The optimal cutoff values were determined from AUROC curve analysis using the Bayesian analysis with the optimal criterion and avoiding the misclassification of true positive patients. Positive predictive value (PPV = true positive cases/all positive cases), negative predictive value (NPV = true negative cases/all negative cases), and diagnostic accuracy (sum of true positive and true negative cases/total number of cases) were calculated. 

This retrospective study was approved by the Local Ethics Committee, and the use of the database conformed to the legislation on personal data protection.

## 3. Results

### 3.1. Patients’ Characteristics

The study included 101 subjects with liver cirrhosis and PTV: 73 men and 28 women, with a mean age of 62.7 ± 9.2 years. The main characteristics of the study group are summarized in Table 1.

### 3.2. Child–Pugh and Meld Classification

According to Child–Pugh classification, 29.2% (7/24) of the subjects with cirrhosis and without HCC were classified as Child–Pugh A, 41.6% (10/24) were Child–Pugh B, and 29.2% (7/24) were Child–Pugh C.

A total of 19.5% (15/77) of the subjects with cirrhosis and HCC were classified as Child–Pugh A, 49.3% (38/77) were Child–Pugh B, and 31.2% (24/77) were Child–Pugh C.

The mean value of the MELD score for subjects with cirrhosis and without HCC was 15.9 ± 7.1, while for those with cirrhosis and HCC, it was 15.8 ± 7.1 (*p* = 0.9616).

### 3.3. Portal Hypertension and PVT Extension

The mean longitudinal spleen diameter in subjects without HCC (*n* = 24) was 16 ± 3.3 cm, and 83.3% (20/24) of them had splenomegaly (spleen size > 12 cm). In subjects with HCC (*n* = 77), the mean spleen diameter was 14.2 ± 2.7 cm, and 76.6% (59/77) of them had splenomegaly (*p* = 0.0085). The mean longitudinal spleen diameter in subjects with TIV (*n* = 75) was 13.9 ± 3.1 cm, while in those with benign PVT (*n* = 26) it was 16.2 ± 3 cm (*p* = 0.0015).

In 84/101 subjects, upper digestive endoscopy was performed. The distribution of subjects according to endoscopic signs of portal hypertension (esophageal and gastric varices) is summarized in Table 2.

The distribution of subjects according to the PVT extension is summarized in Table 3. No significant correlation was found between the extension of PVT and the presence and severity of EV/GV (*p* = 0.515), nor between the extension of PVT and the size of the spleen (*p* = 0.719).

### 3.4. AFP Serum Levels for PVT Characterization

Serum AFP levels were recorded in 81 (80.2%) subjects, 20/81 (24.7%) without HCC, and 61/81 (75.3%) with HCC. In 33.3% (27/81) of subjects, AFP serum levels were <20 ng/dL, in 18.5% (15/81) subjects they were ≥20 and <200 ng/dL, while in 48.2% (39/81) they were ≥200 ng/dL. Mean AFP values were significantly higher in patients with HCC than in those without (*p* < 0.0001). Using CT/MRI as reference, in the subgroup of subjects with AFP serum levels < 20 ng/dL (*n* = 27), PVT was benign in 77.8% (21/27) of cases, and TIV was present in 22.2% (96/27) of them. In the subgroup of subjects with AFP serum levels ≥ 20 and <200 ng/dL (*n* = 15), TIV was present in 73.3% (11/15) of cases, and PVT was benign in 26.7% (4/15) of them, while in subjects with AFP serum levels ≥ 200 ng/dL (*n* = 39), 100% (39/39) were TIV. 

Applying the rule-out AFP cutoff values for HCC (AFP < 20 ng/dL) for the classification of PVT, 78% (21/27) of the subjects were correctly classified as having benign PVT, while when the rule-in cutoff value was used (AFP ≥ 200 ng/dL), 100% (39/39) of the subjects were correctly classified as TIV.

A significant correlation was found between AFP levels and PVT characterization at CEUS (r = 0.28, *p* = 0.0098) and at CT/MRI (r = 0.3, *p* = 0.0057), while no significant correlation was found between AFP levels and the extension of PVT (*p* = 0.4458).

### 3.5. The Performance of CEUS for the Characterization of PVT

According to CEUS results, 74.3% (75/101) of subjects had TIV, while using reference methods (CE-CT/MRI), 72.3% of the subjects had TIV. A more detailed analysis of PVT characterization using both CEUS and the gold standard methods is summarized in Table 4.

A significant correlation was found between the HCC size and PVT extension (r = 0.24, *p* = 0.0318) and between PVT extension and the type of HCC (single, multicentric or diffuse) (r = 0.33, *p* = 0.003). In the subgroup of subjects with benign PVT, 2/5 (40%) had multicentric HCC, while in the subgroup of subjects with TIV, 15/72 (20.8%) subjects had a single lesion smaller than 50 mm, 5/72 (6.9%) had a single lesion larger than 50 mm, 40/72 (55.6%) had multicentric HCC and 12/72 (16.7%) had diffuse HCC. 

The overall performance of CEUS for the characterization of PVT showed 98.6% Se (95%CI 92.6–100%) and 89.3% Sp (95%CI 71.8–97.7%).

The performance of CEUS for the characterization of PVT and the distribution of correctly classified subjects according to the grade of the PVT is summarized in Table 5. A total of 97% (98/101) of subjects included were correctly characterized by CEUS as having benign PVT or TIV, using CE-CT/MRI as the reference method. No significant differences were found between the proportion of subjects correctly classified in the group of subjects with HCC compared with those without HCC: 97.4% (75/77) vs. 95.8% (23/24) *p* = 0.7757.

### 3.6. The Performance of the Combined Use of AFP Serum Levels and CEUS for the Characterization of PVT

We performed a univariate regression analysis to test the association between the characterization of PVT by CEUS and AFP serum levels (ng/dL) with the reference method (CE-CT/MRI). We found that both were significantly associated with the reference method (*p* < 0.001 and *p* = 0.006, respectively). In multivariate logistic regression, the model including PVT characterized by CEUS and AFP serum levels showed that both parameters were associated with PVT characterization by CT/MRI: PVT by CEUS (β = 0.88 ± 0.05, *p* < 0.001) and AFP serum levels (β = 0.16 ∗ 10–4 ± 0.0000026, *p* < 0.03).

Including these factors in multiple regression analysis, we devised a score for classifying PVT: 0.88 ∗ 1 (if the PVT was classified as malignant by CEUS) or * 0 (if the PVT was classified as benign by CEUS) + 0.16 ∗ 10^−4^ ∗ AFP serum levels (ng/dL). 

The score’s optimal cutoff value for predicting malignant PVT was >0.92 (AUROC = 0.99, Se—98.21% [95%CI 90.4–100%], Sp—100% [95%CI 86.3–100%], PPV—1000%, NPV—96.2%, +LR—8.18, −LR—0.018) (Figure 3).

Based on the AUROC comparison, the PVT score performed better than CEUS (AUC—0.99 vs. AUC—0.93, *p* = 0.025) or AFP serum levels (AUC—0.99 vs. AUC—0.96, *p* = 0.047) for characterizing PVT (Figure 4).

## 4. Discussion

HCC is the sixth most prevalent cancer worldwide, one of the most common complications of HCC being PVT. Detection and characterization of PVT are vital in HCC patients because malignant PVT categorizes the patient in an advanced stage (C) in the BCLC staging system, with fewer therapeutic possibilities and contraindication for most curative treatment options [15]. The treatment of HCC patients with PVT is based on the patients’ liver function, the stage of the hepatic lesion, and the extent of PVT [26]. A Chinese expert consensus established the therapeutics options according to the Child–Pugh stage, to the extension of PVT, and the ECOG Performance Status grade, with a variety of available options: surgery, hepatic artery infusion chemotherapy (HAIC), transcatheter arterial chemoembolization (TACE), external beam radiation therapy, internal radiation therapy, local ablation therapies, or systemic therapy [26]. Hence the need for a correct diagnosis and staging of PVT.

Contrast-enhanced ultrasound has emerged in the last fifteen years and proved to be a useful tool, especially for liver applications, such as focal liver lesions and PVT characterization, enhancing the diagnostic value of B-mode and Doppler ultrasound [11]. In the present study, the performance of CEUS for PVT characterization was evaluated, demonstrating 98.6% sensitivity and 89.3% specificity. Other studies evaluated CEUS for PVT characterization with very good results. In a study including 50 patients with HCC and PVT, CEUS had 100% sensitivity and 83% specificity, using CE-CT/MRI as reference [27]. In another study by Tarantino et al., the sensitivities of color Doppler US (CDUS), CEUS, and fine-needle biopsy (FNB) in detecting malignant thrombi in 54 patients with cirrhosis and HCC and PVT were compared, the conclusion being that CEUS had the highest sensitivity (88%) [28]. Moreover, Rossi et al. compared the performance of CEUS and CT for the detection and characterization of PVT, and CEUS had significantly higher sensitivity [29]. In a recently published meta-analysis, the pooled sensitivity and pooled specificity of CEUS in the characterization of PVT were 0.94 (95%CI, 0.89–0.97) and 0.99 (95%CI, 0.80–1.00) [30].

Of the 101 subjects included in our study, CEUS misclassified only two patients, as compared with CT and MRI. This aspect can be attributed to the method’s limitations, mainly the same as for standard US: acoustic window and patient cooperation. Overweight and image artifacts may also affect the results. Furthermore, unlike CT and MRI, CEUS might not be able to detect PVT extension to other splanchnic vessels because of potentially compromised ultrasound access by abdominal gas [30]. 

AFP is a biomarker that has been tested in the diagnostics of HCC, in the surveillance of patients at risk for developing HCC, and for follow-up after treatment. Numerous cutoffs were studied, and the diagnostic accuracy of AFP in small HCC was substantially limited as a surveillance test, its performance being poor. However, when used as a diagnostic test for HCC, values higher than 20 ng/mL showed a quite good sensitivity of 60%, with 90.6% specificity, while for values higher than 200 ng/mL, the sensitivity decreased to 22% but with a higher specificity of 99.4% [31]. The superiority of 200 ng/mL used as a cutoff for HCC smaller than 5 cm was proved in a study by Tateishi et al., where the sensitivity, specificity, and LR+ using 20 ng/mL vs. 200 ng/mL, were 0.49 to 0.71, 0.49 to 0.86, and 1.28 to 4.03, respectively, and from 0.04 to 0.31, 0.76 to 1.0, and 1.13 to 54.25 [32].

Another issue related to AFP and its diagnostic and prognostic value is the occurrence of HCC with low or normal AFP values. Carr et al. observed that 58% of HCC patients in their cohort had AFP values lower than 100 IU/mL (121 ng/dL). Furthermore, 49% of patients with large HCCs (≥5.0 cm) also had normal or low range AFP (≤100 IU/mL) [18]. However, in the same paper, 19% of patients with PVT had AFP <20 IU/mL (24.2 ng/dL), and in 25% of them, AFP ranged from 20 IU/mL to 100 IU/mL, but there was no mention regarding the nature of PVT as benign or TIV.

A few published studies have evaluated the relevance of AFP levels for the characterization of PVT. In the present study, we evaluated the performance of different AFP cutoff levels for PVT classification using CT/MRI as the reference and the correlation between the level of AFP and the type of PVT (benign or TIV) and the extension of PVT. Using a rule-out AFP cutoff value of <20 ng/dL, 78% of the included subjects were correctly classified as having benign PVT, while using a rule-in AFP cutoff value of ≥200 ng/dL, 100% of the subjects were correctly classified as having TIV. Therefore, in the clinical setting of a patient with liver cirrhosis and PVT, AFP levels higher than 20 ng/dL can be highly suggestive of the presence of TIV. Moreover, a significant correlation was found between AFP levels and PVT characterization using all imaging techniques (CEUS, CE-CT/MRI). Similar to our results, in a recently published study that retrospectively evaluated a large cohort of 819 nontransplant HCC patients, it was demonstrated that higher AFP levels are significantly associated with TIV [33]. 

Using multiple regression analysis, we calculated a PVT score that combines CEUS and AFP levels for PVT characterization. The PVT score had a very good performance for predicting malignant TIV, with an AUROC of 0.99, 98.2% Se, and 100% Sp. Moreover, the PVT score performed significantly better when compared with CEUS or AFP levels alone. As far as we know, there are no other published data combining CEUS and AFP for PVT characterization.

Regarding the etiology of liver disease, in our study, the two leading causes of liver cirrhosis associated with PVT were alcohol (ALD)—37.5%, and hepatitis C virus (HCV)—25%. According to a published study, ALD and hepatitis B virus (HBV)-related cirrhosis were found to be the most frequent causes of PVT [34]. However, this correlation was not demonstrated in another study [35]. 

The severity of cirrhosis is correlated with the prevalence of PVT; for compensated cirrhosis, it is as low as 1% [6] and can be 8–25% in candidates for liver transplantation [5]. In patients with cirrhosis and HCC, the prevalence of PVT is higher, approximately 35% [36]. In our study, there were no significant differences in the MELD score between patients with and without HCC, 15.9 ± 7.1 vs. 15.8 ± 7.1, *p* = 0.9616. Also, there were no significant differences in the severity of portal hypertension between subjects with benign PVT and those with TIV.

The limitations of our study are associated with its retrospective, single-center nature that limited the number of patients, and the relatively small sample size of HCC patients. Not all patients in the study group had a recorded AFP level, which further decreased the number of patients, thus precluding us from dividing them according to the etiology of cirrhosis. A possible limitation can also be the lack of an interoperator reproducibility evaluation, knowing that CEUS is an operator-dependent method, like all ultrasound-based methods. In the absence of a histological gold standard, the radiologists’ skills can influence the quality of the reference method (CT and RMN). Furthermore, a second radiological interpretation can increase the confidence in the results. Another limitation of our study can be related to the fact that we did not use machine-learning classifiers or computer-assisted diagnostic tools to assess the diagnostic performance of the parameters we evaluated. We need future prospective multicentric studies to confirm our results with a larger patient group and also to confirm the proposed PVT score performance on other cohorts of patients.

## 5. Conclusions

CEUS is a reliable imaging method for the characterization of PVT, with 98.6% sensitivity and 89.3% specificity for differentiating between benign PVT from TIV. The combination of CEUS and AFP improves the diagnostic performance of PVT characterization (benign vs. malignant).

## Figures and Tables

**Figure 1 diagnostics-12-01145-f001:**
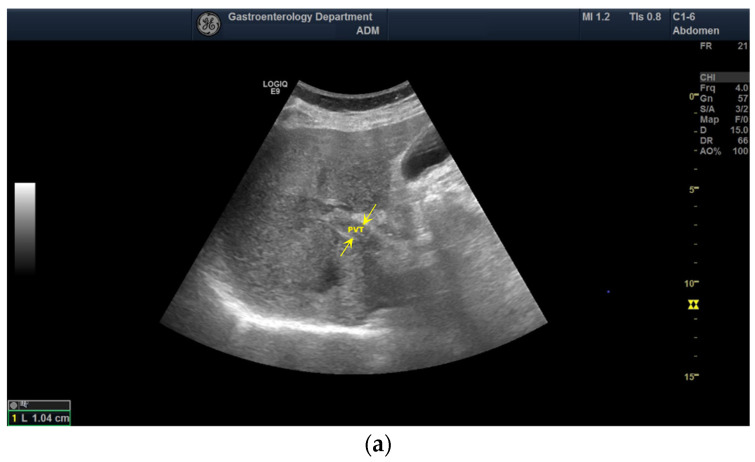
Ultrasound and CEUS aspect of benign PVT: (**a**) aspect in the standard US (between arrows); (**b**) no enhancement in the arterial phase (between arrows); (**c**) no enhancement in the portal phase (between arrows); (**d**) no enhancement in the late phase (between arrows).

**Figure 2 diagnostics-12-01145-f002:**
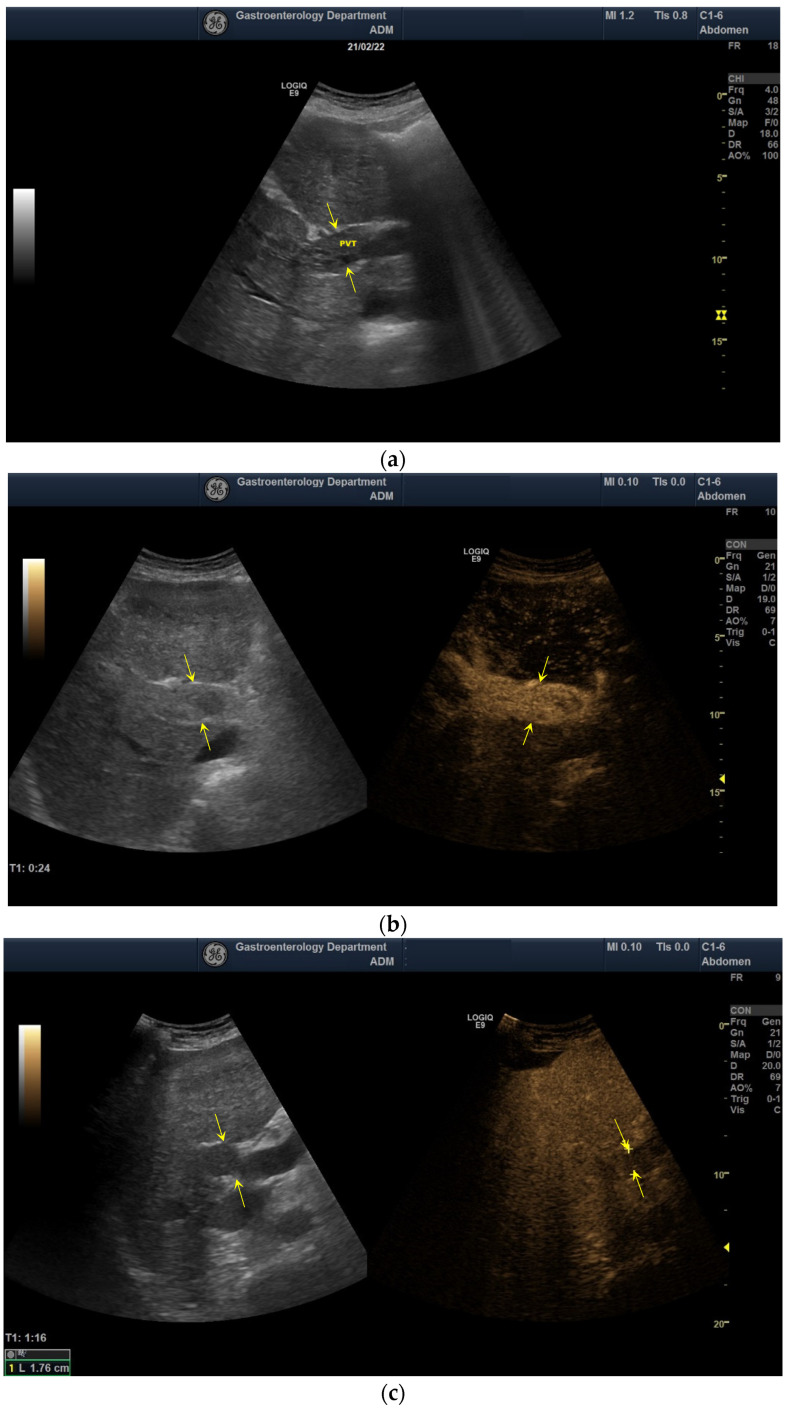
Ultrasound and CEUS aspect of TIV: (**a**) aspect in the standard US (between arrows), (**b**) hyperenhancement in the arterial phase (between arrows), (**c**) washout in the portal phase (between arrows), (**d**) washout in the late phase (between arrows).

**Figure 3 diagnostics-12-01145-f003:**
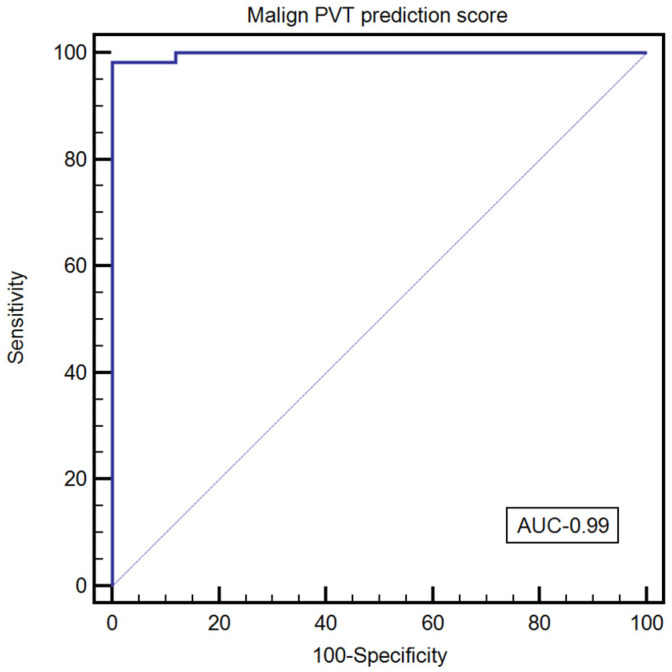
The performance of PVT score for the characterization of PVT.

**Figure 4 diagnostics-12-01145-f004:**
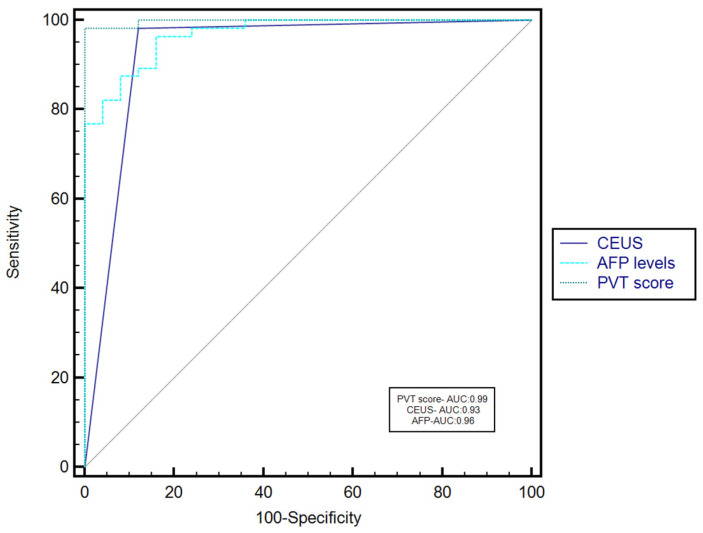
Comparison between receiver operating characteristics for PVT score versus CEUS and AFP serum levels.

**Table 1 diagnostics-12-01145-t001:** Main characteristics of the study group.

Parameter	*n* = 101
Age (years)	62.7 ± 9.2
**Gender**	
Men	73 (72.3%)
Women	28 (27.7%)
**Etiology of PVT:** Liver cirrhosis with HCC	77(76.2%)
Liver cirrhosis without HCC	24 (23.8%)
**Etiology of liver cirrhosis in patients with HCC**	*n* = 77
HCV	27 (35.1%)
HBV	15 (19.5%)
HBV + HDV	2 (2.6%)
HBV + HCV	2 (2.6%)
ALD	16 (20.7%)
ALD + HCV	5 (6.5%)
ALD + HBV	1 (1.3%)
NAFLD	2 (2.6%)
Cryptogenic	7 (9.1%)
**Etiology of liver cirrhosis in patients without HCC**	*n* = 24
AH	1 (4.2%)
HCV	6 (25%)
PBC	1 (4.2%)
SSC	1 (4.2%)
ALD	9 (37.5%)
HBV	2 (8.3%)
HBV + HDV	2 (8.3%)
HBV + ALD	1 (4.2%)
NAFLD	1 (4.2%)

Data are presented as number and percentage or mean ± standard deviation; *n*—number; AH—autoimmune hepatitis, HCV—hepatitis C virus; PBC—primary biliary cirrhosis; SSC-HBV—hepatitis B virus; HDV—hepatitis D virus, ALD—alcohol-related liver disease; NAFLD—nonalcoholic fatty liver disease; PBC—primary biliary cirrhosis; HCC—hepatocellular carcinoma, AH—autoimmune hepatitis.

**Table 2 diagnostics-12-01145-t002:** The distribution of subjects according to the presence of endoscopic signs of portal hypertension.

	**Subjects without HCC** ***n* = 22**	**Subjects with HCC** ***n* = 62**	***p*-Value**
No or small EV (grade 1)	7 (31.8%)	19 (30.6%)	0.8698
Grade 2 and 3 EV	11 (50%)	30 (48.4%)	0.9051
EV and GV	4 (18.2%)	13 (21%)	0.9777
	**Subjects with TIV** ***n*= 61**	**Subjects with Benign PVT** ***n* = 23**	***p*-Value**
No or small EV (grade 1)	20 (32.8%)	6 (26.1%)	0.7532
Grade 2 and 3 EV	31 (50.8%)	10 (43.5%)	0.7248
EV and GV	10 (16.4%)	7 (30.4%)	0.2629

*n*—number, HCC—hepatocellular carcinoma, EV—esophageal varices, GV—gastric varices.

**Table 3 diagnostics-12-01145-t003:** The distribution of subjects according to PVT extension.

	Grade 1	Grade 2	Grade 3	Grade 4
Patients with HCC *n* = 77	31 (40.3%)	35 (45.4)	5 (6.5%)	6 (7.8%)
Patients without HCC *n* = 24	9 (37.5%)	8 (33.3%)	5 (20.8%)	2 (8.4%)

*n*—number, HCC—hepatocellular carcinoma, PVT—portal vein thrombosis.

**Table 4 diagnostics-12-01145-t004:** PVT characterization, using CEUS and the gold standard methods.

Subjects	CEUS	CT/MRI	*p*-Value
Subjects with HCC *n* = 77	72 (93.5%) TIV 5 (6.5%) benign PVT	72 (93.5%) TIV 5 (6.5%) benign PVT	0.7438 0.7438
Subjects without HCC *n* = 24	3 (12.5%) TIV 21 (87.5%) benign PVT	1 (4.2%) TIV 23 (95.8%) benign PVT	0.6047 0.6047

*n*—number, HCC—hepatocellular carcinoma, PVT—portal vein thrombosis, CEUS—Contrast-enhanced ultrasound, CT—computed tomography, MRI—magnetic resonance imaging.

**Table 5 diagnostics-12-01145-t005:** Performance of CEUS for the characterization of PVT and distribution of correctly classified subjects according to PVT grade.

PVT Grade	CEUS Performance				Correctly Classified Subjects (%)
	Sensitivity	Se 95%CI	Specificity	Sp 95%CI	
I	100%	87.7–100%	83.3%	51.6–97.9%	95% (38/40)
II	100%	89.7–100%	100%	66.4–100%	100% (43/43)
III	100%	47.8–100%	80%	28.4–99.5%	100% (10/10)
IV	83.3%	35.9–99.6	100%	15.8–100%	87.5% (7/8)

PVT—portal vein thrombosis, CEUS—contrast-enhanced ultrasound, 95%CI—95% confidence interval.

## Data Availability

The data underlying the findings of the study are available on request to the corresponding author (e-mail address: roxanasirli@gmail.com).

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
