# Peer review of "Usefulness of Imaging and Biological Tools for the Characterization of Portal Vein Thrombosis in Hepatocellular Carcinoma"

_diagnostics, 2022, doi:10.3390/diagnostics12051145_

Round 1
Reviewer 1 Report
In this study, the authors evaluated the performance of CEUS for the characterization of portal vein thrombosis in patients with and without HCC.
- Please spell Se and Sp in the abstract and at their first appearance in the text.
- Please specify how the gold standard diagnosis (on CT_/MRI) was made
- A comment on the role of contrast-enhanced US/EUS (e.g., mural nodule in IPMN) should be included. Doing so, cite PMID: 34217751
Author Response
- Please spell Se and Sp in the abstract and at their first appearance in the text.
Response: Thank you for your suggestion, we made the corrections accordingly (page 1, line 16).
- Please specify how the gold standard diagnosis (on CT_/MRI) was made.
Response: Thank you. We added details of the CE-CT and CE-MRI scanning protocols in the M&M section (page 6, line 155).
- A comment on the role of contrast-enhanced US/EUS (e.g., mural nodule in IPMN) should be included. Doing so, cite PMID: 34217751.
Response: Thank you for the suggestion. We found the mentioned paper interesting and valuable and added it to the references section (page 2, line 57); (page 14, line 476).
Reviewer 2 Report
Malignant portal vein thrombosis is a contraindication for liver transplantation. Efforts towards promising point-of-care diagnosis has been emerging.
=Major comments:
- The utility of CEUS in diagnosing PVT has been one of the already known facts in the published, relevant reports and guidelines over the past decade
- Therefore, novelty is only recognized if advanced or combined applications of CEUS, significant statistics, and in-depth insights can be demonstrated
- Combined use of CEUS and AFP is novel to date. However, this is not statistically available or discussed in the current report. Is it feasible to remodel the current report?
- Liver fibrosis stages, AFP, CEUS, cirrhosis, HCC, benign PVT, and malignant PVT are dependent and confound each other in statistics from clinical experiences
- The authors are well-experienced and the authors know this filed very well. However, results delineating the independent significance adjusting for the aforementioned confounding factors (and/ or demonstrating the diagnostic performances through machine-learning classifiers if adequate datasets are available) are lacking in the current analysis of the current dataset with limitations in the sample size and statistics
=Miscellaneous:
- The current submission with a guideline-changing potential proposing combined utility of CEUS and AFP as predictors of PVT (benign—performances: ; malignant—performances: …if feasible when reformed) is well written in Introduction section. However, only univariate, separate, non-robust, non-combined correlations were derived from the current invaluable dataset
- At least present the numerically improved diagnostic performances combining the diagnostic tools currently studied
- It would be invalid to address a spleen diameter is equivalent to portal hypertension. A body of spleen-based indices are available based on the studies from your institutes and recommended by the guidelines
- The distinct correlations between AFP and malignant/ benign PVT are unclear and inadequate in both the statistic results demonstrated and the supporting discussions
(in published studies and [APASL] guidelines, the mechanisms, citations, recommendations for low levels of precancerous AFP and high thresholds for HCC were clearly discussed)
- “two AFP levels cut-offs: 20 ng/ml and 200 ng/ml”—on the basis of which published studies or guidelines?
- Issues on AFP were not well cited, utilized, or discussed
- “Meld”, “EASL”, “EFSUMB”?
- It would be more appropriate than the present to spell out Se and Sp in Abstract section
- As regulated by STARD (STAndards for the Reporting of Diagnostic accuracy studies) Checklist, performances including +LR are typically addressed, compared, and used for recommendation evaluations of published evidence. The current performances need to be expanded if feasible to enhance the merit. The current report is not qualified as judged using the checklist
- More limitations can be spotted throughout the entire report and need to be discussed
- Key images are interesting but lacking in the current report
Author Response
Dear Reviewer,
Thank you very much for your comments and suggestions for our manuscript diagnostics-1624685, entitled " Usefulness of Imaging and Biological Tools for the Characterization of Portal Vein Thrombosis in Hepatocellular Carcinoma", which are very valuable and have helped us revise and improve our manuscript. We appreciate all of your work on our manuscript.
We have carefully studied your comments and corrected our manuscript accordingly. We hope this version of the manuscript meets your approval. All changes have been highlighted in the manuscript. The corrections in the paper and the response to the editor and reviewers are as follows:
- Malignant portal vein thrombosis is a contraindication for liver transplantation. Efforts towards promising point-of-care diagnosis has been emerging.
Response: Thank you for your comment, we added corresponding completions (page 1, line 40).
- The utility of CEUS in diagnosing PVT has been one of the already known facts in the published, relevant reports and guidelines over the past decade. Therefore, novelty is only recognized if advanced or combined applications of CEUS, significant statistics, and in-depth insights can be demonstrated.
Response: Thank you for your comment and observation. We agree with the reviewer. The value of CEUS for PVT diagnosing and characterization has been previously reported. However, the combined use of CEUS with AFP serum levels is of high interest and has superior clinical utility. We performed an analysis of the performance of the two methods (CEUS and AFP), both independently and together in a score to predict the presence of malignant PVT. The combined PVT score had a superior performance for PVT characterization. Changes have been added and highlighted in the manuscript. (page 10, line 300)
- Combined use of CEUS and AFP is novel to date. However, this is not statistically available or discussed in the current report. Is it feasible to remodel the current report?
Response: Thank you for your comment. As mentioned previously, we combined AFP and CEUS into a score for PVT characterization, using multiple regression analysis. The PVT score had the highest AUROC, sensitivity, and specificity when compared to CEUS or AFP. Changes were made accordingly. (page 11, line 308)
- Liver fibrosis stages, AFP, CEUS, cirrhosis, HCC, benign PVT, and malignant PVT are dependent and confound each other in statistics from clinical experiences.
Response: Thank you for your comment. We performed univariate and multivariate logistic regression analysis which showed that AFP and CEUS were associated with PVT characterization as benign or malignant in our study. These findings were added to the manuscript. (page 11, line 317)
- The authors are well-experienced and the authors know this field very well. However, results delineating the independent significance adjusting for the aforementioned confounding factors (and/ or demonstrating the diagnostic performances through machine-learning classifiers if adequate datasets are available) are lacking in the current analysis of the current dataset with limitations in the sample size and statistics.
Response: Thank you for your observation. We agree with the reviewer. Our study has some important limitations, which have been discussed in more depth now. However, the statistics have been improved in the revised version, the combined performance for CEUS and AFP has been added and compared to CEUS and AFP alone. (page 7, line 198)
- The current submission with a guideline-changing potential proposing combined utility of CEUS and AFP as predictors of PVT (benign—performances: ; malignant—performances: …if feasible when reformed) is well written in Introduction section. However, only univariate, separate, non-robust, non-combined correlations were derived from the current invaluable dataset.
Response: Thank you for your comment. A prediction score, using both CEUS and AFP serum levels, for predicting the presence of malign PVT was formulated. The comparison between receiver operating characteristics for PVT score versus CEUS and AFP serum levels was also performed. Changes were added and highlighted in the manuscript. (page 10, line 298)
- At least present the numerically improved diagnostic performances combining the diagnostic tools currently studied.
Response: Thank you for your comment, we added corresponding completions. (page 11, line 311 ).
- It would be invalid to address a spleen diameter is equivalent to portal hypertension. A body of spleen-based indices are available based on the studies from your institutes and recommended by the guidelines
Response: Thank you for the comment. We did not intend to address the fact that the diameter of the spleen is equivalent to portal hypertension, but the relationship between portal hypertension and splenomegaly is validated and already demonstrated by previous studies. We can either refer to splenomegaly caused by portal congestion or due to tissue hyperplasia and fibrosis or to the fact that the increase in spleen size itself is followed by an increase in splenic blood flow, which participates in portal hypertension, the relationship between the two entities is undeniable. In the present paper, only a concise analysis of the spleen diameter of the included subjects based on different conditions (the presence/absence of HCC, benign/malign PVT) was performed without the intention to establish a direct relationship between portal hypertension and spleen diameter.
- The distinct correlations between AFP and malignant/ benign PVT are unclear and inadequate in both the statistic results demonstrated and the supporting discussions (in published studies and [APASL] guidelines, the mechanisms, citations, recommendations for low levels of precancerous AFP, and high thresholds for HCC were clearly discussed)
Response: Thank you for your comment. The statistics regarding AFP value have been improved and the related discussions as well. (page 2, line 63 ); (page 12, line 363 ); (page 13, line 387 )
- “two AFP levels cut-offs: 20 ng/ml and 200 ng/ml”—on the basis of which published studies or guidelines?
Response: Thank you for your comment. We used two AFP levels cut-offs for the diagnosis of TIV based on the EASL HCC management guideline: 20 ng/ml and 200 ng/ml. Changes have been made in the manuscript accordingly. (page 3, line 112 )
- Issues on AFP were not well cited, utilized, or discussed
Response: Thank you for your suggestion, we made the corrections accordingly in the discussion and introduction parts. (page 2, line 63);(page 3, line 102);(page 12, line 363).
- “Meld”, “EASL”, “EFSUMB”?
Response: Thank you for your suggestion. We have made proper changes. (page 2, line 90); (page 2, line 93);(page 3, line 112).
- It would be more appropriate than the present to spell out Se and Sp in Abstract section
Response: Thank you for your suggestion, we made the corrections accordingly. (page 1, line 16).
- As regulated by STARD (STAndards for the Reporting of Diagnostic accuracy studies) Checklist, performances including +LR are typically addressed, compared, and used for recommendation evaluations of published evidence. The current performances need to be expanded if feasible to enhance the merit. The current report is not qualified as judged using the checklist
Response: Thank you for your suggestion, we made the corrections accordingly. In its current form, the manuscript meets the checklist criteria according to STARD 2015 for studies of diagnostic accuracy. (page 2, line 85); (page 3, line 102);(page 6, line 155); (page 1, line 198); (page 11, line 312).
- More limitations can be spotted throughout the entire report and need to be discussed
Response: Thank you for your suggestion, the limitation part of our discussions has been improved. (page 13, line 406).
- Key images are interesting but lacking in the current report
Response: Thank you for your suggestions, we added the images according to the reviewer's suggestions. (page 4, line 134).
Round 2
Reviewer 2 Report
Revise all the typographical errors?-"malign" throughout the entire well-revised manuscript.
Author Response
Dear Reviewer,
Thank you very much for your comment and suggestion for our manuscript diagnostics-1624685, entitled " Usefulness of Imaging and Biological Tools for the Characterization of Portal Vein Thrombosis in Hepatocellular Carcinoma". We appreciate all of your work on our manuscript.
- Revise all the typographical errors?-"malign" throughout the entire well-revised manuscript.
Response: Thank you for your comment and your observation. We revised the entire manuscript for typographical errors and made marked up the changes using the “Track Changes” function.